# Delivery of circulating lipoproteins to specific neurons in the Drosophila brain regulates systemic insulin signaling

Marko Brankatschk*, Sebastian Dunst, Linda Nemetschke, Suzanne Eaton*

[1]Department of Molecular, Cell and Developmental Biology, Max Planck Institute for Molecular Cell Biology and Genetics, Dresden, Germany

**Abstract** The Insulin signaling pathway couples growth, development and lifespan to nutritional conditions. Here, we demonstrate a function for the *Drosophila* lipoprotein LTP in conveying information about dietary lipid composition to the brain to regulate Insulin signaling. When yeast lipids are present in the diet, free calcium levels rise in Blood Brain Barrier glial cells. This induces transport of LTP across the Blood Brain Barrier by two LDL receptor-related proteins: LRP1 and Megalin. LTP accumulates on specific neurons that connect to cells that produce Insulin-like peptides, and induces their release into the circulation. This increases systemic Insulin signaling and the rate of larval development on yeast-containing food compared with a plant-based food of similar nutritional content.

## Introduction

Nutrient sensing by the central nervous system is emerging as an important regulator of systemic metabolism in both vertebrates and invertebrates (*Lam et al., 2005*; *Levin et al., 2011*; *Rajan and Perrimon, 2012*; *Bjordal et al., 2014*; *Linneweber et al., 2014*). Little is known about how nutrition-dependent signals pass the blood brain barrier (BBB) to convey this information. Like the vertebrate BBB, the BBB of *Drosophila* forms a tight barrier to passive transport, and is formed by highly conserved molecular components (*Bundgaard and Abbott, 2008*; *Stork et al., 2008*; *Abbott et al., 2010*). Its simple structure and genetic accessibility make it an ideal model to study how nutritional signals are communicated to the CNS. Insulin and Insulin-like growth factors are conserved systemic signals that regulate growth and metabolism in response to nutrition. Although *Drosophila* do not have a single pancreas-like organ, they do produce eight distinct *Drosophila* Insulin/IGF-like peptides (Dilps) that are expressed in different tissues (*Riedel et al., 2011*; *Colombani et al., 2012*; *Garelli et al., 2012*). A set of three Dilps (Dilp2,3,5), released into circulation by Dilp-producing cells (IPCs) in the brain, have particularly important functions in regulating nutrition-dependent growth and sugar metabolism; ablation of IPCs in the CNS causes Diabetes-like phenotypes, slows growth and development, and produces small, long-lived adult flies (*Rulifson et al., 2002*; *Broughton et al., 2005*; *Partridge et al., 2011*). Systemic Insulin/IGF signaling (IIS) increases in response to dietary sugars, proteins and lipids. Sugars act on IPCs directly to promote Dilp release (*Haselton and Fridell, 2010*), but other nutrients are sensed indirectly through signals from the fat body—an organ analogous to vertebrate liver/adipose tissue (*Colombani et al., 2003*; *Geminard et al., 2009*; *Rajan and Perrimon, 2012*).

The *Drosophila* fat body produces two major types of lipoprotein particles: Lipophorin (LPP), the major hemolymph lipid carrier, and Lipid Transfer Particle (LTP). LTP transfers lipids from the intestine to LPP. These lipids include fatty acids from food, as well as from endogenous synthesis in the intestine (*Palm et al., 2012*). LTP also unloads LPP lipids to other cells (*Van Heusden and Law, 1989*;

**\*For correspondence:**
brankats@mpi-cbg.de (MB);
eaton@mpi-cbg.de (SE)

**Competing interests:** The authors declare that no competing interests exist.

**Reviewing editor**: Mani Ramaswami, Trinity College Dublin, Ireland

**eLife digest** How does an animal sense if it is well nourished or not, and then regulate its metabolism appropriately? This process largely relies on the animal's body deciphering signals that that are transmitted between different organs in the form of molecules and hormones. Many animals—ranging from insects to mammals (including humans)—also use their brains to sense and decipher these nutritional signals.

A signaling pathway involving the hormone insulin controls how various different animals grow and develop—and how long they will live—based on these animals' food intake. Insulin is produced in mammals by an organ called the pancreas. But in the fruit fly *Drosophila*, this hormone is produced by cells within different tissues, including the insect's brain.

The fruit fly is used to study many biological processes because it is easy to work with in a laboratory. Insulin-producing cells make and release insulin-like molecules into the insect's hemolymph (a blood-like fluid) in response to sugar and to other nutrients (which are detected via molecules generated in a fruit fly organ called the fat body). The fat body produces lipophorin, a protein which carries fat molecules in the hemolymph, and which is known to be able to move from the hemolymph to the brain and accumulate within the brain. The fat body also produces lipid transfer protein (or LTP), which transfers fats absorbed or made within the insect's gut onto lipophorin, and can also unload fat molecules to other insect cells. If LTP can also enter the brain, and what it might do there, was unclear.

Brankatschk et al.now discover that LTP can cross the 'blood brain barrier' in fruit fly larvae and can accumulate over time on their insulin-producing cells and the neurons in direct contact with these cells. This accumulation depends on the flies' diet: flies fed a diet made from yeast cells accumulated LTP on these neurons, while those fed only on sugar and proteins did not.

Furthermore Brankatschk et al. found that when they switched flies from a yeast-based to a plant-based diet, the larvae grew more slowly and the flies lived longer. Both of the diets contained abundant calories and nutrients, but contained slightly different kinds of fat molecules. The fly larvae on the plant-based diet also accumulated less LTP on their insulin-pathway neurons, and insulin signaling was reduced.

Branskatschk et al. also found that fat molecules from the yeast-based diet activated the cells of the blood brain barrier, and that this encouraged LTP to be transported the brain. Blocking LTP from crossing the blood brain barrier reduced insulin signaling, slowed the growth of the fly larvae, and extended the lifespan of the flies.

These findings of Brankatschk et al. thus reveal that fat-containing molecules carry information about specific nutrients to the brain. The extent to which these mechanisms operate in other animals—such as mammals—remains to be explored.

*Canavoso et al., 2004*; *Parra-Peralbo and Culi, 2011*). LPP crosses the BBB and accumulates throughout the brain. It is required for nutrition-dependent exit of neural stem cells from quiescence (*Brankatschk and Eaton, 2010*). Here, we investigate possible functions of LTP in the brain.

## Results

Immunostaining reveals LTP on specific neurons and glia in larval brains. (*Figure 1A,C–E*, *Figure 1—figure supplement 1–3*, and *Videos 1–4*). First instar brains have on average three LTP-positive neurons per brain lobe, increasing to 13 in early third instar larvae (*Figure 1B*, *Figure 1—figure supplement 1*). We used cell type-specific RNAi to distinguish whether LTP in the brain came from circulation, or whether it was produced in the CNS. Knock-down of *ltp* in the fat body reduces but does not eliminate LTP from circulation (*Figure 1F*). Staining larval brains from these animals for LTP reveals reduced staining on both neurons and glia (*Videos 5 and 6*). To investigate this issue in more detail, we quantified LTP-positive neurons after knock-down of *ltp* in neurons, glia, or fat body. To ensure that we compared larvae of similar developmental stages we quantified glial cell numbers, which increase during larval development (*Figure 1B',G'*). Only fat body-specific *ltp* knock-down reduces neuronal LTP staining in the brain (*Figure 1G*, *Figure 1—figure supplement 3*). Thus, LTP particles secreted by the fat body cross the Blood Brain Barrier and become enriched on specific neurons.

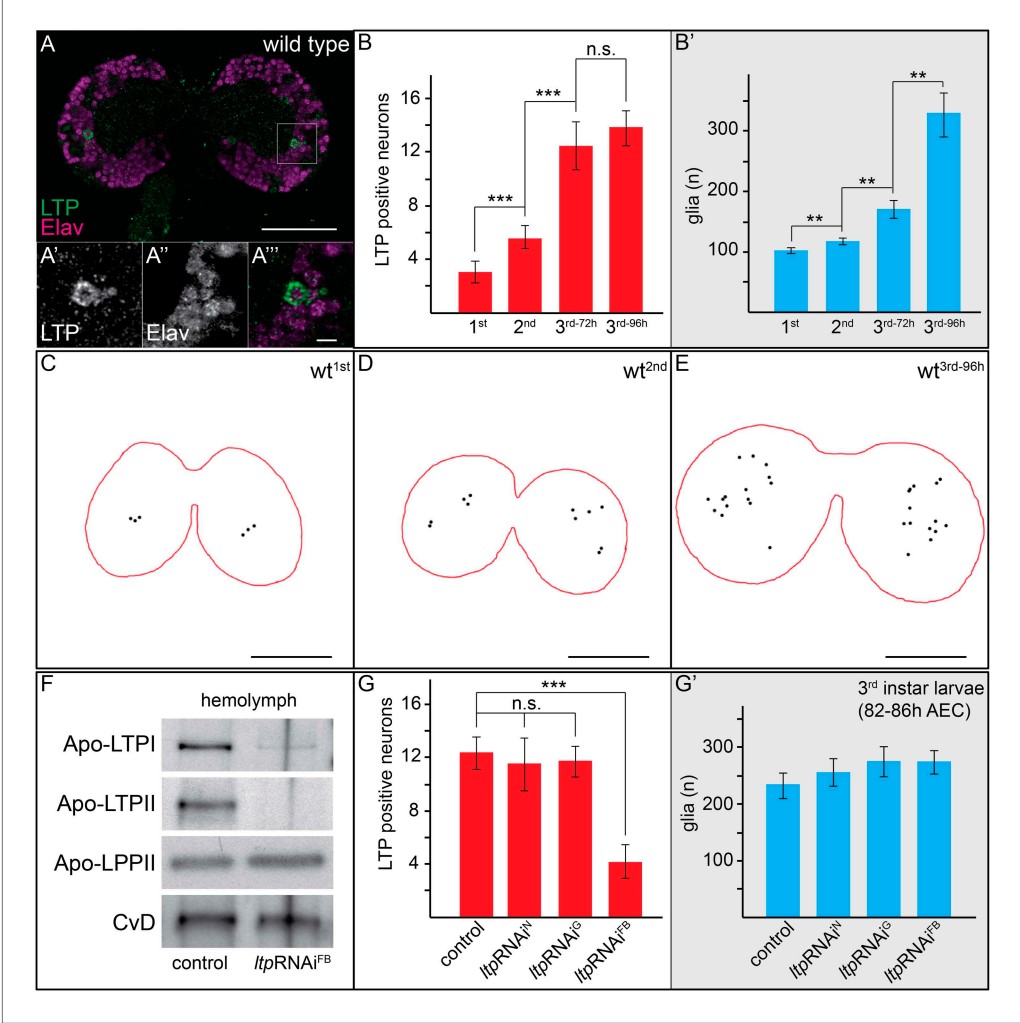

**Figure 1**. Circulating LTP crosses the BBB and accumulates on neurons. (**A**) Confocal section of the CNS from a wt larva reared on YF (wt$^{YF}$), at the level of the big commissure, stained for LTP (green) and Elav (magenta). (**A'–A'''**). Magnified boxed region in (**A**) (**B–B'**) Total numbers of LTP-positive neurons/brain lobe (**B**) or Repo-positive glia/brain (**B'**) of wt$^{YF}$ larvae of different ages, quantified from 50–60 confocal sections per brain. Numbers indicate larval instar; superscripts indicate age in hours after egg collection. (**C–E**) Cartoons depict positions of all LTP-positive neurons (black dots) identified in confocal stack of three wt$^{YF}$ brains of different ages, stained for LTP and Elav. (**F**) Western blot showing equal volumes of hemolymph from control$^{YF}$ larvae, and larvae with FB-specific knock-down of *ltp* (*ltp*RNAi$^{FB}$). Blots are probed for Apo-LTPI, Apo-LTPII, Apo-LPPII and Cv-D. (**G** and **G'**) Total numbers of LTP-positive neurons/brain lobe (**G**) and Repo-positive glia/brain (**G'**) quantified from control$^{YF}$ larvae, and larvae where *ltp* has been knocked-down in neurons (*ltp*RNAi$^{N}$), glia (*ltp*RNAi$^{G}$), or FB (*ltp*RNAi$^{FB}$). Error bars indicate standard deviation. *** = $p < 0.001$, ** = $p < 0.01$, n.s. = not significant (Student's *t* test). Scale bars indicate 50 μm (**A** and **C–E**) or 5 μm (**A'–A'''**).

The following figure supplements are available for figure 1:

**Figure supplement 1**. Numbers of LTP positive neurons rise over larval development.

**Figure supplement 2**. Small glial subsets enrich LTP positive.

**Figure supplement 3**. Circulating LTP crosses the BBB and enriches on neurons.

To understand the CNS functions of LTP, we sought to identify the neurons where it accumulated. We first investigated whether LTP co-localized with Dilp2-positive neurons in early third instar larval brains. Dilp2 is present not only in IPCs, which lie dorso-lateral to the big commissure, but also on

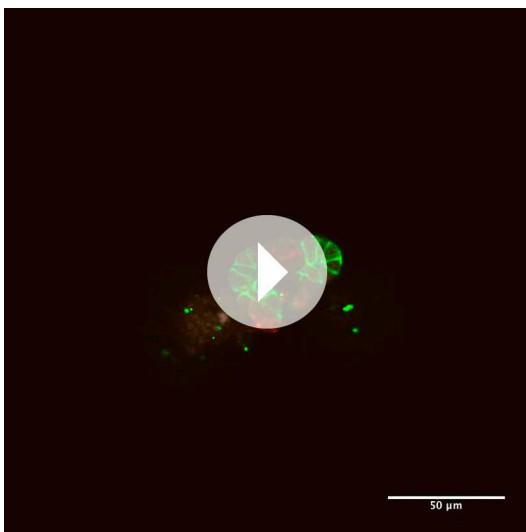

**Video 1**. Confocal stack from wild type first instar larval brain probed for LTP (green), Dilp2 (red) and Repo (grey). Sections are spaced 1.5 μm apart, scale bars indicate 50 μm.

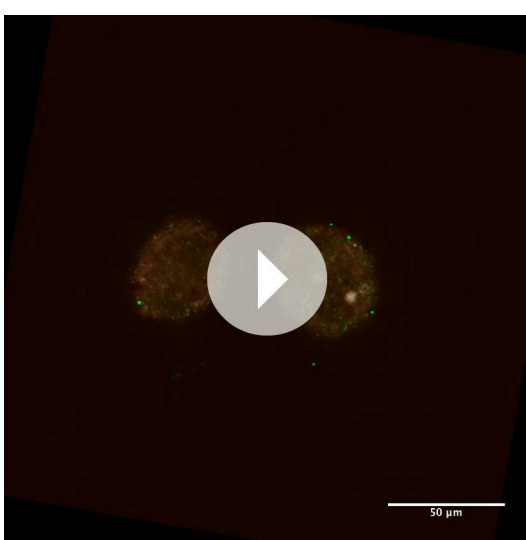

**Video 2**. Confocal stack from wild type second instar larval brain probed for LTP (green), Dilp2 (red) and Repo (grey). Sections are spaced 1.5 μm apart, scale bars indicate 50 μm.

three neurons per hemisphere that do not produce Dilps. These neurons recruit Dilp2 from IPCs using IMPL2, a Dilp2-binding protein (*Bader et al., 2013*). Although each hemisphere contains 7-8 neurons that express IMPL2 at this stage, only 3 of these detectably recruit Dilp2, and 2 of these 3 also accumulate LTP. These neurons can be unambiguously identified by the presence of both LTP and Dilp2, by their position, by lack of expression of *dilp2-GAL4*, and by expression of *impL2-GAL4* (*Figure 2*, *Figure 2—figure supplement 1*). Henceforth, we refer to these neurons as Dilp2-recruiting neurons (DRNs). Starting in the third larval instar, LTP accumulates on two specific DRNs located dorsal to the big commissure (*Figure 2A,C,D*, *Figure 2—figure supplement 2*). Although LTP is found on other neurons at earlier stages, its accumulation on DRNs is developmentally regulated. We also sometimes (6/20 sampled CNS) detect LTP on a subset of IPC neurons (*Figure 2B*). Thus, upon entering the brain, LTP accumulates on IPCs and specific neurons that recruit Dilp2.

Thus far, we have described LTP localization in larvae reared on a rich medium: 'yeast food' (YF) (*Carvalho et al., 2012*). Systemic IIS is high in these animals, as reflected by predominantly cytoplasmic localization of FOXO in fat body cells (*Figure 3E,E'*) (*van der Horst and Burgering, 2007*). To ask whether neuronal LTP accumulation was regulated by nutrition, we asked how it responded to starvation. Late second instar larvae transferred from YF to food containing only glucose (GF) arrest growth and development. Systemic IIS is strongly reduced, as indicated by nuclear accumulation of FOXO in fat body cells (*Figure 3A,A'*). Although LTP still circulates (*Figure 3G*), it accumulates on many fewer neurons in glucose-fed larvae, and is never detected on DRNs (*Figure 3F–F''*). Thus, the presence of glucose is insufficient to allow high systemic IIS or accumulation of LTP on most neurons.

We next examined the role of lipids and proteins in promoting neuronal LTP accumulation and systemic IIS. We compared larvae fed on YF, with those transferred at late second instar to a lipid-depleted food (LDF), which contains a yeast autolysate that has been chloroform-extracted to remove lipid. LDF contains proteins and sugars, and has a similar caloric content to YF. However, larvae arrest on LDF because of the absence of sterols (*Carvalho et al., 2010*), and systemic IIS is low (*Figure 3B,B'*). Brains of LDF-fed larvae contain more LTP-positive neurons than those of glucose-fed larvae, but fewer than YF-reared larvae. In particular, LTP is never detected on DRNs (*Figure 3F–F''*).

To address the sterol-dependence of neuronal LTP accumulation, we supplemented LDF with 10 μM cholesterol (LDSF). Larvae transferred from YF to LDSF grow slowly and give rise to viable adults of reduced size (*Carvalho et al., 2010*). Consistent with this, nuclear FOXO staining in the fat body indicates that systemic IIS is low (*Figure 3C,C'*; *Carvalho et al., 2010*). Sterol addition further

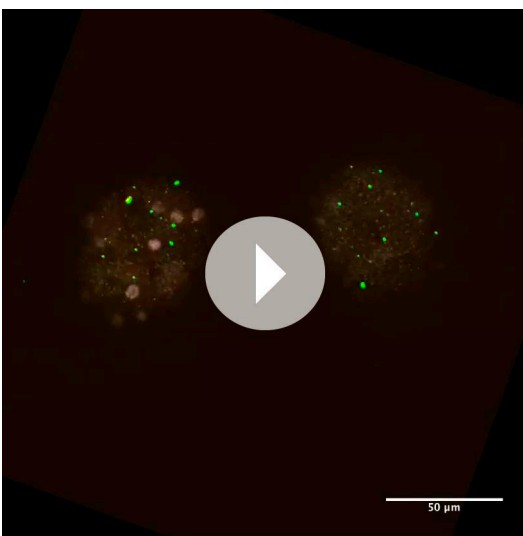

**Video 3**. Confocal stack from wild type third instar larval brain probed for LTP (green), Dilp2 (red) and Repo (grey). Sections are spaced 1.5 µm apart, scale bars indicate 50 µm.

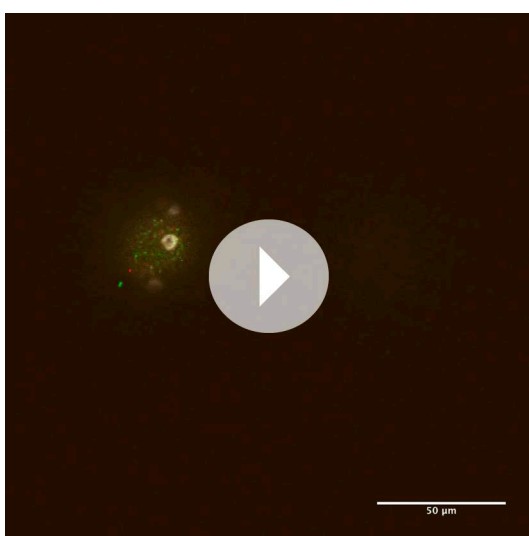

**Video 4**. Confocal stack from wild type third instar larval brain probed for Elav (red), Repo (grey), LTP (green) and DAPI (blue). Sections are spaced 1.5 µm apart, scale bars indicates 50 µm.

increases the number of LTP-positive neurons, but still does not allow LTP accumulation on DRNs (*Figure 3F–F''*, and *Figure 3—figure supplement 1*). Thus, a component present in yeast food, but not the chloroform-extracted yeast autolysate, is required for accumulation of LTP on DRNs, and for high-level systemic IIS.

We wondered whether lipids in general were required for LTP accumulation on DRNs. Interestingly however, experiments with plant food (PF) suggest that bulk lipid is not sufficient. PF contains no yeast and is based entirely on plant materials. It has the same caloric content as YF, and slightly more lipid with a different fatty acid composition (*Carvalho et al., 2012*, see 'Materials and methods'). Surprisingly, LTP is found only occasionally on DRNs in larvae transferred from YF to PF. Depletion of LTP on DRNs occurs within 16 hr of transfer and is reversible in the same time frame (*Figure 3F–F''* and *Figure 4A,A',C* and *Figure 3—figure supplement 2*). Strikingly, despite the abundant calories derived from carbohydrates, proteins and lipids, feeding with PF specifically slows the larval growth phase without lengthening embryonic or pupal development. PF also dramatically extends adult lifespan compared to YF (*Figure 4E–G*). This suggests that systemic IIS is reduced when larvae are fed with PF, compared to YF. Indeed, FOXO is predominantly nuclear in PF-reared larvae (*Figure 3D,D'*).

To investigate mechanisms underlying the differences in systemic IIS in larvae fed with YF and PF, we examined the pathway at different levels. We first focused on the neuronal activity of the IPCs. Neuronal activity correlates with higher levels of intracellular free calcium, which can be detected using the GCaMP reporter (*Reiff et al., 2005*). This GFP reporter increases its fluorescence in response to free calcium. We expressed the GCaMP construct either in IPC neurons (under the control of *dilp2-GAL4*) or in all neurons (under the control of *rab3-GAL4*) and compared GFP fluorescence in brains from YF and PF-reared larvae. While the activity of most neurons is not affected by feeding with these different foods (*Figure 5A,B*), the activity of the IPCs was dramatically higher on YF than on PF (*Figure 5C,D*; *Videos 7 and 8*). We quantified the number of IPCs exhibiting detectable fluorescence of the GCaMP reporter in 5 brains each of larvae fed with PF or YF. On YF, we detected activity of the GCaMP reporter in 4.8 ± 0.6 neurons per brain lobe, while on PF only 1 ± 1.3 neurons per brain lobe exhibited detectable reporter fluorescence. To ask whether higher neuronal activity of IPCs resulted in elevated release of Dilp2 into the hemolymph, we probed Western blots of hemolymph from YF and PF-reared larvae with antibodies to Dilp2. YF-reared animals have higher levels of circulating Dilp2 than PF-reared animals (*Figure 5E*). Thus, IPCs release more Dilp2 when animals are fed with YF. We next examined different molecular readouts of IIS in different tissues of PF and YF-reared larvae. YF increases PI3K activity in salivary glands (*Figure 5F,G*), as revealed by membrane localization

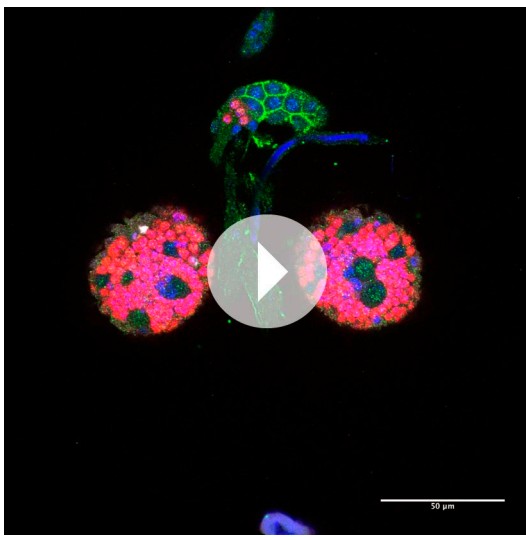

**Video 5**. Confocal stack from UAS:ltpRNAi/+ third instar larval brain probed for Repo (grey), Dilp2 (red) and LTP (green). Sections are spaced 1.5 µm apart, scale bars indicate 50 µm.

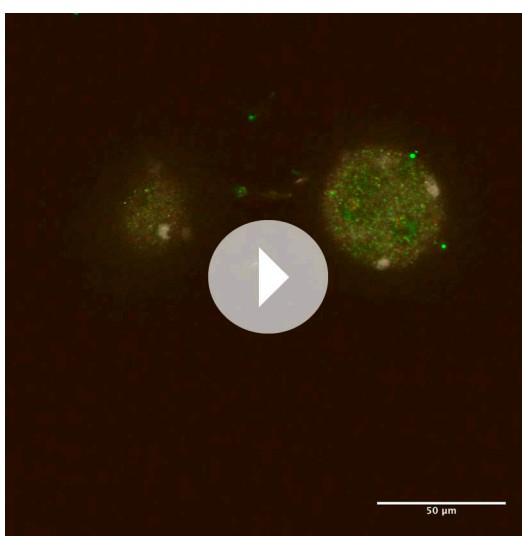

**Video 6**. Confocal stack from lpp-Gal4>UAS:ltpRNAi third instar larval brain probed for Repo (grey), Dilp2 (red) and LTP (green). Sections are spaced 1.5 µm apart, scale bars indicate 50 µm.

of the PH<sup>GFP</sup> reporter construct. Furthermore, phosphoAKT levels are higher in fat bodies of YF-reared larvae (*Figure 5H*). Thus, feeding with YF activates IPCs, causing them to secrete more Dilp2 into circulation, thereby increasing systemic IIS.

To examine whether the differences between plant and yeast lipids were responsible for changes in IPC activity, we prepared chloroform extracts of PF and YF and used them to supplement LDF. Yeast lipids supported faster larval growth compared to plant lipids (*Figure 5I*). Furthermore, the GCaMP reporter reveals that IPC neurons are only active when larvae are fed with yeast lipids (*Figure 5J,K*; *Videos 7 and 8*). Thus, IPC neurons respond differently to the types of lipids present in YF and PF.

How does YF promote IPC activity and Dilp release? We wondered whether the YF-dependent localization of LTP to the DRNs might be responsible. To investigate this, we first asked whether IMPL2-expressing neurons actually contact the IPCs. We labeled the projections of IMPL2 expressing neurons by driving a transmembrane Cherry (Cherry) under the control of *impL2-Gal4*, and stained larval brains for both Dilp2 and IMPL2. A subset of Cherry-labeled projections, also containing IMPL2, extends from the DRNs towards the IPCs (*Figure 6B′,B″,B‴*). Dilp2 staining reveals a subset of projections from the IPCs extending towards the DRNs (*Figure 6A*—class 2 projections and *Figure 6B,B″*). They meet in a region where IMPL2 begins to colocalize with Dilp2 (*Figure 6B‴*). Thus, it is likely that DRNs and IPCs communicate with each other directly.

How does the activity of IMPL2-positive neurons influence that of IPCs? To address this question, we inhibited synaptic release by knocking down Rabs 3 and 27—two Rabs that both contribute to synaptic vesicle release (*Mahoney et al., 2006*; *Pavlos et al., 2010*). To do this, we used homologous recombination to generate N-terminal YFP fusions of Rab3 and Rab27 at their endogenous chromosomal loci. This renders them sensitive to knock-down by anti-GFP RNAi (*Figure 6—figure supplement 1*). Rabs 3 and 27 are expressed almost exclusively in neurons (manuscript in preparation). Thus, even for GAL4 drivers that are active in many tissues, this approach will affect only the neurons that express the GAL4 driver. When anti-GFP RNAi is driven in the IPCs in a background homozygous for both YFPRab3 and YFPRab27, larvae exhibit the slow growth phenotype characteristic of reduced IIS (*Figure 6C*). Reducing levels of Rab3/Rab27 in the IMPL2-expressing neurons slows larval growth (*Figure 6C*), and reduces phosphorylation of AKT in the fat body (*Figure 5H*). This suggests that signals sent by IMPL2-expressing neurons promote systemic IIS.

Since LTP localizes to DRNs only on yeast food, we wondered whether it was required to promote systemic Insulin signaling. We therefore asked whether IIS in YF-reared larvae depended on LTP. RNAi-mediated *ltp* knock-down in the fat body causes larval arrest in the second or third instar

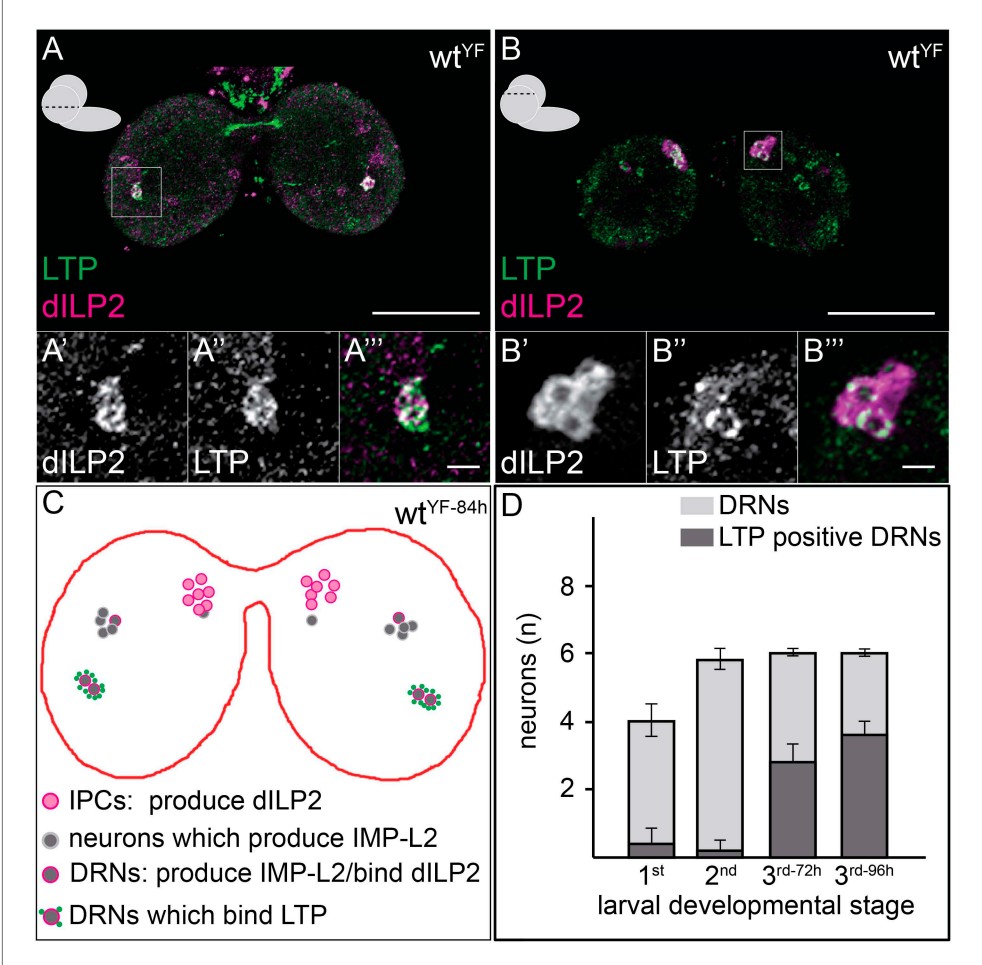

**Figure 2**. LTP accumulates on neurons positive for Dilp2 and IMPL2. Confocal sections at the level of (**A–A'''**) or dorsal to (**B–B'''**) the big commissure from third instar wt[YF] larval brains stained for Dilp2 (magenta) and LTP (green). Boxed regions show LTP/Dilp2 double-positive neurons. Scale bar = 50 µm (**A** and **B**) or 5 µm (**A'''** and **B'''**). (**C**) Cartoon drawn from a single sectioned brain showing positions of IPCs (magenta), IMPL2-producing cells that recruit Dilp2 (DRNs) (grey with magenta rim), and IMPL2-producing cells negative for Dilp2 (grey). LTP (green dots) is found on a subset of DRNs. (**D**) shows average number of neurons double-positive for Dilp2/IMPL2 (DRNs, grey) and the number of these that are LTP-positive (black) in larval brains of different stages. Error bars = standard deviation.

The following figure supplements are available for figure 2:

**Figure supplement 1**. A small subset of Dilp2 positive neurons expresses IMPL2.

**Figure supplement 2**. In the last larval developmental stage DRNs enrich LTP.

---

(**Palm et al., 2012**). Fat body cells contain lower levels of phosphoAKT (**Figure 5H**) and predominantly nuclear FOXO (**Figure 7K,K'**). Thus LTP is required for high-level IIS in the fat body on YF.

Reduced IIS in *ltp* mutants might reflect a requirement for LTP on DRNs. Alternatively, loss of LTP in other tissues might affect Insulin signaling indirectly. Blocking transport of LTP across the BBB in YF-reared larvae could distinguish these possibilities. We therefore examined requirements for different LDL receptor family proteins in BBB transport of LTP. LTP localizes normally to DRNs and other neurons in larvae double mutant for LDL receptor homologues LpR1 and LpR2 (**Figure 7**, and **Figure 7—figure supplement 1–3**; **Khaliullina et al., 2009**). Thus, LpR1 and LpR2 are not dominant transporters of LTP across the BBB. However, we discovered redundant functions for LRP1 and LRP2

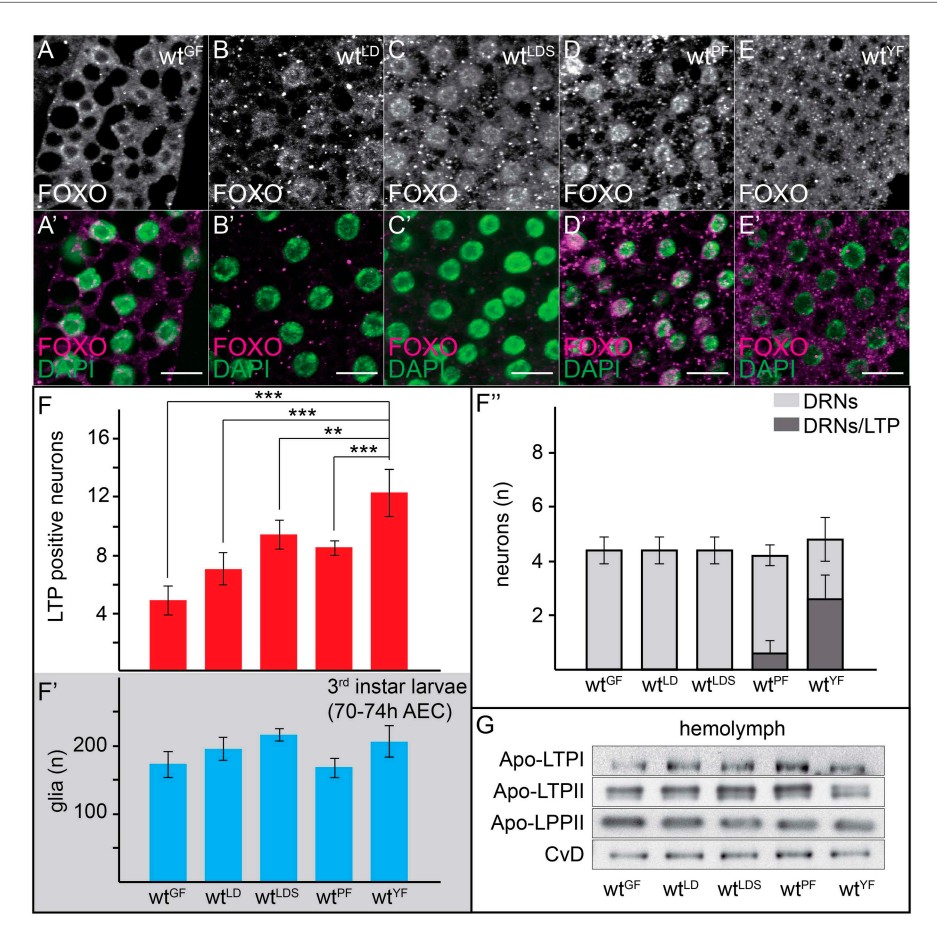

**Figure 3**. Neuronal LTP accumulation is diet-dependent. (**A–E'**) Confocal stack projections of fat bodies from early third instar larvae raised on GF (**A** and **A'**), LDF (**B** and **B'**), LDSF (**C** and **C'**), PF (**D** and **D'**) and YF (**E** and **E'**) probed for FOXO (**A–E** and **A–E'** magenta) and DAPI (**A'–E'**, green). Scale bars = 20 µm. (**F–F'''**) average number of LTP-positive neurons/brain lobe (**F**), glial cells/brain (**F'**) and LTP-positive DRNs (**F''**) in brains of larvae transferred from YF to indicated diets in the late second instar. Error bars indicate standard deviation. T-test significance: **p < 0.01, ***p < 0.001. (**G**) Equal hemolymph volumes from wt larvae raised on indicated food sources Western blotted and probed for the indicated Apolipoproteins.

The following figure supplements are available for figure 3:

**Figure supplement 1**. Only yeast food promotes LTP enrichments on DRNs.

**Figure supplement 2**. LTP enrichments on DRNs are reversible.

---

in BBB transport of LTP. Both proteins are predominantly expressed in glial cells (**Figure 7A–F**). Double mutants die as third instar larvae (not shown). Although LTP circulates at normal levels (**Figure 7— figure supplement 1B**), the number of LTP-positive neurons is reduced (**Figure 7—figure supplement 1A,A'**). A similar phenotype is produced by glial-specific double knockdown of LRP1/2, which halves the number of LTP-positive neurons, and completely blocks localization to DRNs. Knockdown of either LRP alone produces intermediate phenotypes, suggesting they function redundantly (**Figure 7G–G'''**, **Figure 7—figure supplement 3**). Thus, both LRPs transport LTP across the BBB to DRNs. Transport of LTP to other neurons has a less strict requirement for LRP1/2.

If the absence of LTP on DRNs reduces systemic IIS, then losing LRP1/2 in BBB glia should reproduce this effect. Indeed, glial-specific LRP1/2 knock-down slows larval growth and delays pupariation, and emerging adults live longer—all phenotypes suggesting reduced IIS (**Figure 7H–I,M**). Consistent with this, levels of circulating Dilp2 are lower in these animals (**Figure 5E**), AKT is less phosphorylated

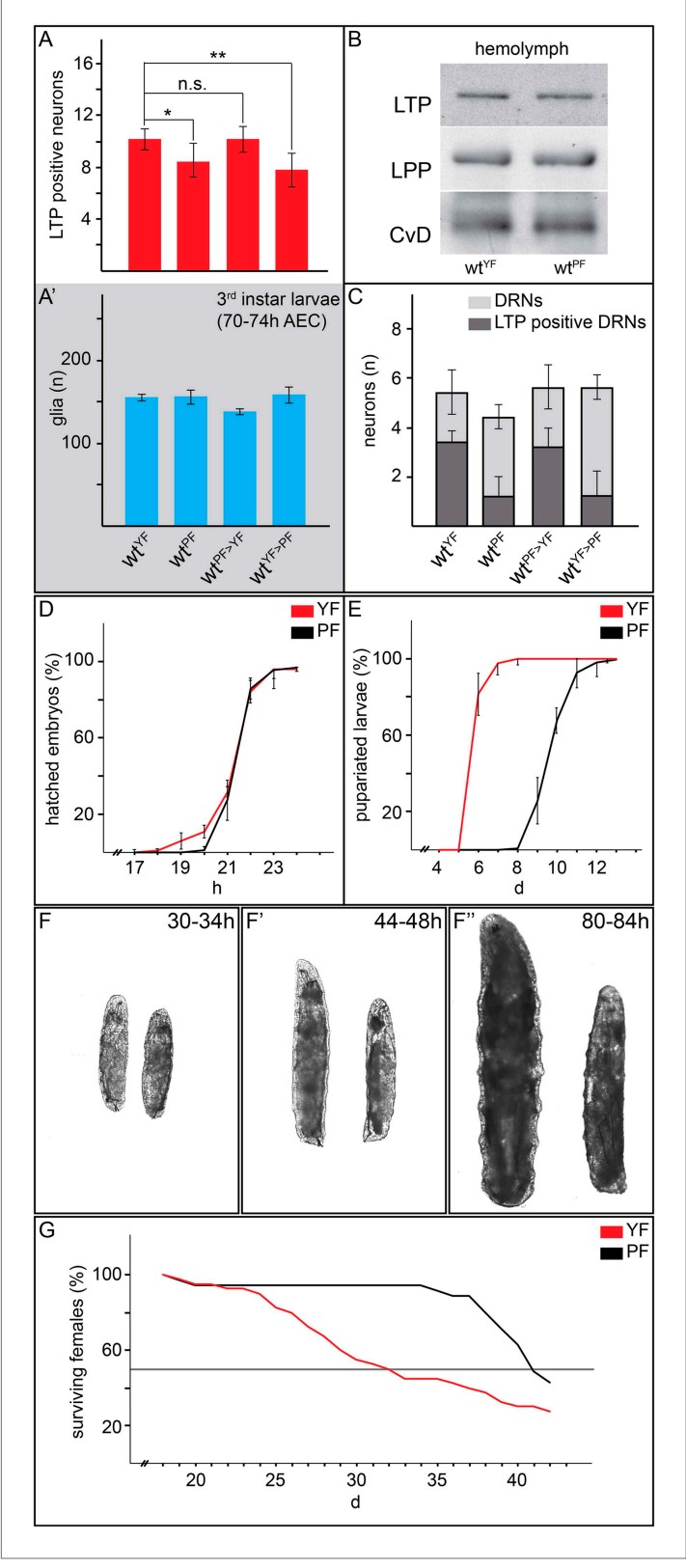

**Figure 4**. Yeast food promotes fast larval development but reduces average life span. (**A**, **A'**, **C**) Charts depict LTP positive neurons/brain hemisphere (**A**) and Repo positive glia/brain (**A'**), and DRNs or LTP positive DRNs/brain (**C**) from larvae reared on YF, PF, raised on PF till late second instar and transferred for 16hr on YF, and raised on YF
*Figure 4. Continued on next page*

*Figure 4. Continued*

till late second instar and transferred for 16 hr on PF. p-values (Student's *t*test) are indicated, * = p < 0.05, ** = p < 0.01, n.s. = not significant, error bars = standard deviation. (**B**) Equal volumes hemolymph from early third instar wt larvae reared on YF or PF probed for Apo-LTP, Apo-LPP and Cvd. (**D, E, G**) Plotted are percentages (Y-axis) of hatched embryos (**D**; $n^{YF}$ = 198, $n^{PF}$ = 198; parental flies kept for three generations on respective food types before embryo collection), pupariated larvae (**E**; $n^{YF}$ = 163 and $n^{PF}$ = 133; time of pupal development was unchanged) and of living mated females (**G**; $n^{YF}$ = 40 and $n^{PF}$ = 35) over time (X-axis). Please note, $LD_{50}$ indicated with grey line (**G**). (**F–F″**) Exemplary photographs of staged wt larvae bred on YF (left) or PF (right). Hours after egg collection (AEC) are indicated in top right corner.

(*Figure 5H*), and FOXO is predominantly nuclear in larval fat body cells (*Figure 7L,L'*). These data support the idea that blocking transport of LTP to DRNs reduces the release of Dilp2 by IPCs, thereby lowering systemic IIS.

What mechanisms promote the BBB transport of LTP to DRNs on yeast food? We first wondered whether levels of LRP1,2 might change. However, we observed no obvious differences in LRP1,2 staining in the brain on YF vs PF (data not shown). However we were surprised to observe dramatically higher levels of free intracellular calcium in BBB glia when larvae are fed YF compared to PF. Driving the free intracellular calcium sensor GCaMP using *repo-GAL4* (which is active in all glia) reveals a marked reduction in GFP fluorescence in BBB glia relative to other glia. We confirmed this result by specifically driving the reporter in BBB glia using *moody-GAL4* (*Figure 8A–D*).

To investigate whether Ca++ signaling in BBB glia was sufficient to promote transport of LTP to DRNs, we asked whether ectopically inducing Ca++ influx would allow LTP to accumulate on DRNs even when larvae were fed with PF. To do this, we drove the expression of TRPA1 with *moody-GAL4* in PF-reared larvae and stained their brains for LTP and Dilp2. Strikingly, while LTP is undetectable on DRNs of wild type larvae reared on PF, it accumulates to high levels on these neurons when Ca++ influx is induced by TRPA1 expression in BBB cells (*Figure 8E,E'*). Accumulation is specific to DRNs and neuronal LTP localization in these larvae precisely mimics its localization on YF. Furthermore, Western blotting shows that phosphorylation of AKT in the fat body increases compared to wild type larvae fed with PF (*Figure 8G*). Taken together, these data show that YF increases free cytoplasmic Ca++ in BBB glia. This is sufficient for the specific transport of LTP to DRNs and elevated systemic IIS. Interestingly, the accumulation of LTP on other neurons does not appear to require elevated Ca++ in BBB cells.

## Discussion

In summary, this work demonstrates a key requirement for lipoproteins in conveying nutritional information across the BBB to specific neurons in the brain. As particles that carry both endogenously synthesized and diet-derived lipids, lipoproteins are well-positioned to perform this function. Our data suggest that transport of LTP across the BBB to DRNs influences communication between DRNs and the Dilp-producing IPCs, increasing the release of Dilp2 into circulation. Since the IPCs also deliver Dilp2 to the DRNs, this indicates that these two neuronal populations may communicate bidirectionally. How might LTP affect the function of DRNs? One possibility is that it acts to deliver a signaling lipid to the DRNs. It could do so either directly, or indirectly by promoting lipid transfer from LPP, which is present throughout the brain (*Brankatschk and Eaton, 2010*). LTP enrichment on specific neurons may increase lipid transfer to these cells.

This work highlights a key function for BBB cells in transmitting nutritional information to neurons within the brain. Feeding with yeast food increases free calcium in BBB glia, which then increases transport of LTP to DRNs. How might BBB cells detect the difference between yeast and plant food? Our data suggest differences in the lipid composition of yeast and plant-derived foods are responsible. Our previous work has shown that the lipids in these foods differ in their fatty acid composition. Yeast food has shorter and more saturated fatty acids than plant food (*Carvalho et al., 2012*). How could these nutritional lipids affect the activity of BBB glia? Interestingly, differences in food fatty acid composition are directly reflected in the fatty acids present in membrane lipids of all larval tissues including the brain (*Carvalho et al., 2012*). Thus, it is possible that the bulk membrane properties of BBB glia are different on these two diets. Membrane lipid composition is known to affect a variety of signaling events (*Simons and Toomre, 2000*; *Lingwood and Simons, 2010*). Alternatively, yeast food may influence the specific fatty acids present in signaling lipids that activate BBB glia.

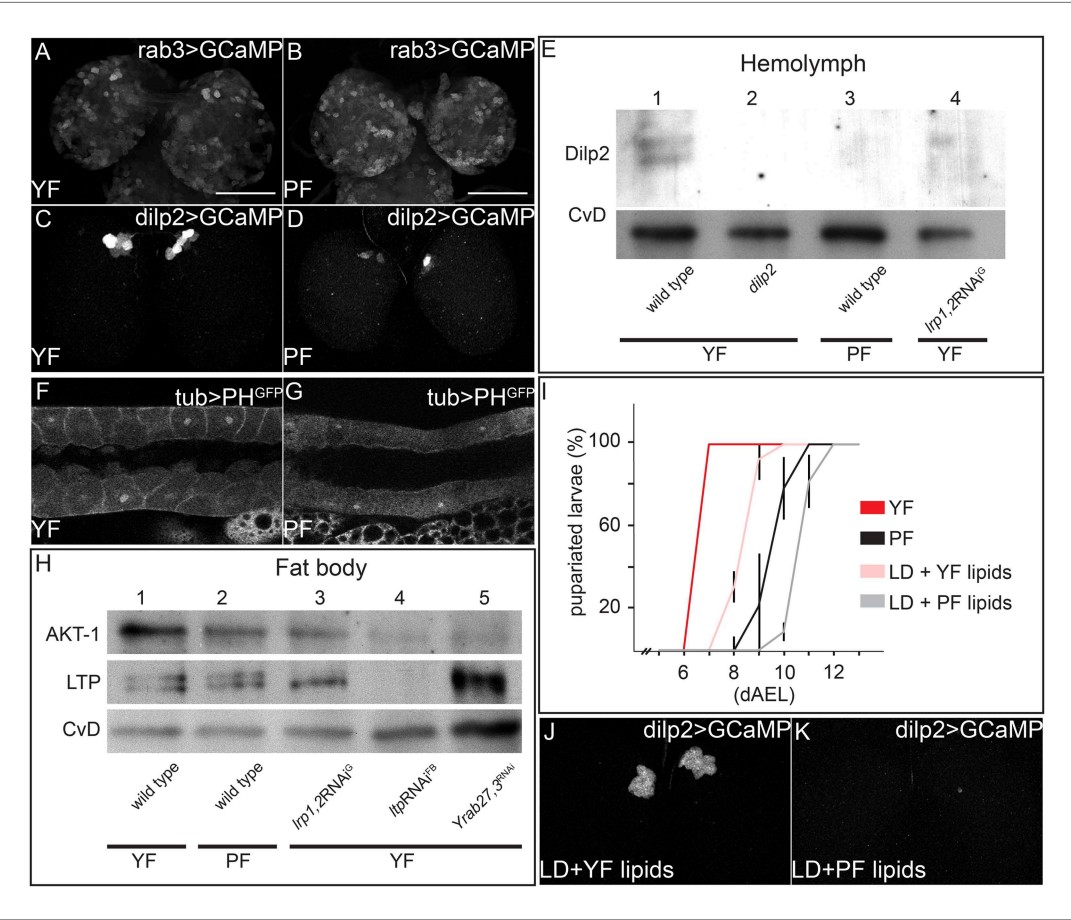

**Figure 5**. Yeast food components activate IPCs to release Dilp2 and induce systemic Insulin signaling. (**A–D**) show GFP fluorescence from collapsed confocal stacks of whole brains expressing GCaMP under the control of *rab3-GAL4* (**A** and **B**) or *dilp2-GAL4* (**C** and **D**) from larvae reared on YF (**A** and **C**) or PF (**B** and **D**). (**E**) Hemolymph blots probed for Dilp2 and CvD from wild type (lanes 1 and 3), *dilp2* mutant (lane 2) and *repo-GAL4>lrp1,2*RNAi (lane 4) larvae reared on yeast food (YF) or plant food (PF) as indicated. (**F** and **G**) show single confocal sections of salivary glands expressing the PIP$_3$ reporter PH$^{GFP}$ under the direct control of the tubulin promoter. Larvae were reared on YF (**F**) or PF (**G**). (**H**) Western blots from fat body lysates probed for phospho-AKT, LTP and CvD as indicated. Lysates are from wild type (lanes 1, 2), *repo-GAL4>lrp1,2*RNAi (lane 3), *lpp-GAL4>ltp*RNAi (lane4) and *YFP-rab27;YFP-rab3;impl2-GAL4>gfp*RNAi (lane 5). Larvae were reared on yeast food (YF) or plant food (PF) as indicated. (**I**) shows percentages of larvae that have pupariated at different days after egg laying (dAEL). Larvae were reared on yeast food (red, n = 49), on plant food (black, n = 34), or on lipid-depleted food supplemented with either yeast lipids (light red, n = 23) or plant lipids (grey, n = 24). (**J** and **K**) show GFP fluorescence from collapsed confocal stacks of whole brains expressing GCaMP under the control of *dilp2-GAL4* from larvae fed on lipid-depleted food supplemented with yeast lipids (**J**) or lipid-depleted food supplemented with plant lipids (**K**). See also *Videos 7 and 8*.

We demonstrate an unexpected functional specialization of the BBB glial network, which permits specific and regulated LTP transport to particular neurons. How this specificity arises is an important question for the future. We note that a subset of glial cells within the brain also accumulates LTP derived from the fat body. Could these represent specific transport routes from the BBB? An alternative possibility is that transport depends on neuronal activity. Mammalian LRP1 promotes localized transfer of IGF in response to neuronal activity (***Nishijima et al., 2010***). Could LTP delivery by LRP1 and LRP2 in the *Drosophila* brain depend on similar mechanisms? The remarkable specificity of LTP trafficking in the *Drosophila* CNS provides a novel framework for understanding information flow between the circulation and the brain.

To what extent might this be relevant to vertebrate systems? While it is clear that the vertebrate brain (unlike that of *Drosophila*) does not depend on lipoproteins to supply it with bulk sterols

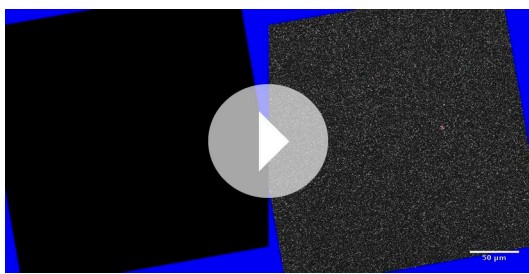

**Video 7**. Confocal stack from dilp2-Gal4>UAS:GCaMP larval brain reared on PF, right panel shows the same images at higher gain. Images show GFP fluorescence of the GCaMP reporter. Scale bars indicate 50 μm.

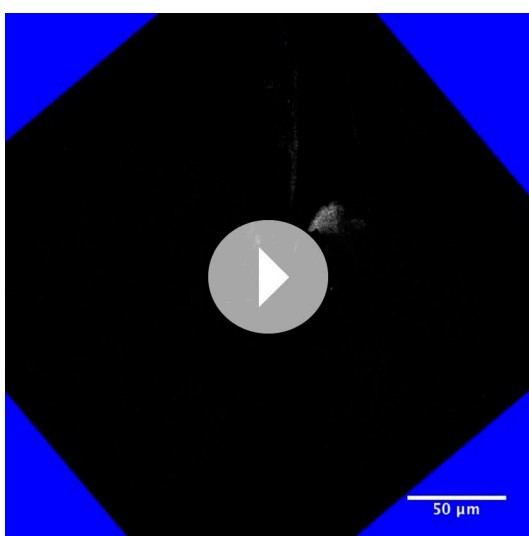

**Video 8**. Confocal stack from dilp2-Gal4>UAS:GCaMP larvae reared on YF. Images show GFP fluorescence of the GCaMP reporter. Scale bars indicate 50 μm.

(*Orth and Bellosta, 2012*), this does not rule out possible functions for these particles in nutrient sensing. The vertebrate cerebrospinal fluid is rich in many types of HDL particles, including those containing ApoA-1, which is not expressed in the brain—this suggests that at least some lipoprotein particles in the brain may derive from the circulation (*Wang and Eckel, 2014*). Consistent with this idea, ApoA-I can target albumin-containing nanoparticles across the BBB in rodents (*Zensi et al., 2010*). Recent work suggests that lipoproteins may be the source of specific Long Chain Fatty Acids that signal to the hypothalamus to regulate glucose homeostasis, since neuronal lipoprotein lipase is required for this process (*Wang et al., 2011*; *Wang and Eckel, 2014*). Thus, it would be interesting to investigate whether circulating mammalian lipoproteins might reach a subset of neurons in the hypothalamus.

It has been known for some time that increasing the amount of yeast in the diet of lab grown *Drosophila melanogaster* increases the rate of development and adult fertility, but reduces lifespan (*Sang, 1949*; *Leroi et al., 1994*; *Mair et al., 2005*). Here, we show that flies have evolved specific mechanisms to increase systemic IIS in response to yeast, independently of the number of calories in the diet or its proportions of sugars proteins and fats. What pressures could have driven the evolution of such mechanisms? In the wild, *Drosophila melanogaster* feed on rotting plant material and their diets comprise both fungal and plant components. *Drosophila* disperse yeasts and transfer them to breeding sites during oviposition improving the nutritional resources available to developing larvae (*Markow and O'grady, 2008*). Yeast that are able to induce more rapid development of the agents that disperse them may propagate more efficiently. On the other hand, it has been noted that *Drosophila* species that feed on ephemeral nutrient sources like yeasts or flowers have more rapid rates of development (*Markow and O'grady, 2008*) than other species. It may be that, even within a single species, the ability to adjust developmental rate to the presence of a short-lived resource is advantageous. Humans subsist on diets of both plant and animal materials that during most of evolution have differed in their availability. It would be interesting to investigate whether Insulin/IGF signaling in humans might respond to qualitative differences in the lipid composition of these nutritional components.

## Materials and methods

### Cloning

#### pCM43:rab3-Gal4
A DNA fragment containing 5.4 kbp 5' to the *rab3* ATG start codon was fused to a *gal4* cDNA and inserted into the multiple cloning site of pCM43 (gift from B Dickson), placing it upstream of a SV40 polyadenylation sequence and a mini-white gene.

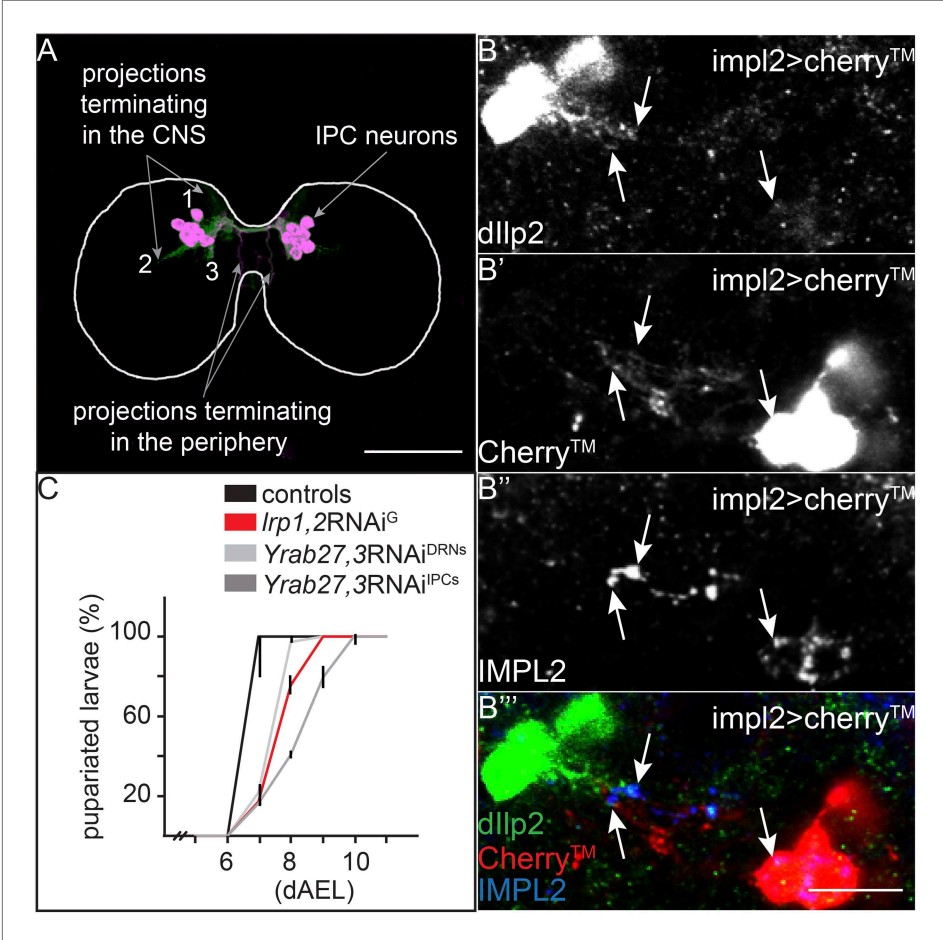

**Figure 6**. IPC and IMPL2-positive neurons are in direct contact. (**A**) Projected confocal stacks of an early third instar brain expressing transmembrane Cherry under the control of *dilp2-GAL4,* stained for Cherry (green) and Dilp2 (magenta). Most Dilp2 is detected in projections that terminate in the periphery (as determined from examining individual sections). However, Cherry staining reveals other projections that enter the central neuropil (***Linneweber et al., 2014***) or remain in other regions of the CNS (***Levin et al., 2011***; ***Bjordal et al., 2014***). (**B**) Shows a single confocal section from *impl2-GAL4>cherry*TM larval brains probed for Dilp2 (**B**, green in **B'''**), Cherry (**B'**, red in **B'''**) and IMPL2 (**B''**, blue in **B'''**). White arrows point to colocalization between Dilp2 and IMPL2. Scale bars indicate 50 μm (**A**) or 10 μm (**B–B'''**). (**C**) Shows percentages of yeast food-reared larvae of different genotypes that have pupariated at different days after egg laying (dAEL). Black indicates pooled results from two control genotypes: UAS:*lrp1,2*RNAi/+ and UAS:*gfp*RNAi/+ (n = 81). Red indicates *repo-GAL4>lrp1,2*RNAi (n = 109). Light grey indicates *YFP-rab27;YFP-rab3;impl2-GAL4>gfp*RNAi (n = 76). Dark grey indicates *YFP-rab27;YFP-rab3;dilp2-GAL4>gfp*RNAi (n = 120).

The following figure supplement is available for figure 6:

**Figure supplement 1**. *Y-rab3* and *Yrab27* are reduced by *gfp*RNAi.

## UAS:membrane tagged mcherry

An 840 bp fragment containing the predicted signal sequence of *lipophorin* was fused to a HA-epitope triplet and cloned 5′ to an *mcherry* cDNA. This was fused at its 3′ end to a CD8 transmembrane domain sequence that was flanked at the 5′ end with the MYC-epitope and the 3′ end with a V5 epitope. The construct was cloned into pCM43 vector containing a 5xUAS-*hsp70* promoter element followed by a multiple cloning site, a SV40 element and mini-*white* gene.

### Immunohistochemistry

Larval brains were dissected on ice in Graces medium and fixed with 4%PFA at room temperature. Fat bodies were dissected in 4%PFA-Graces medium at room temperature. Samples were stained in 7.5%

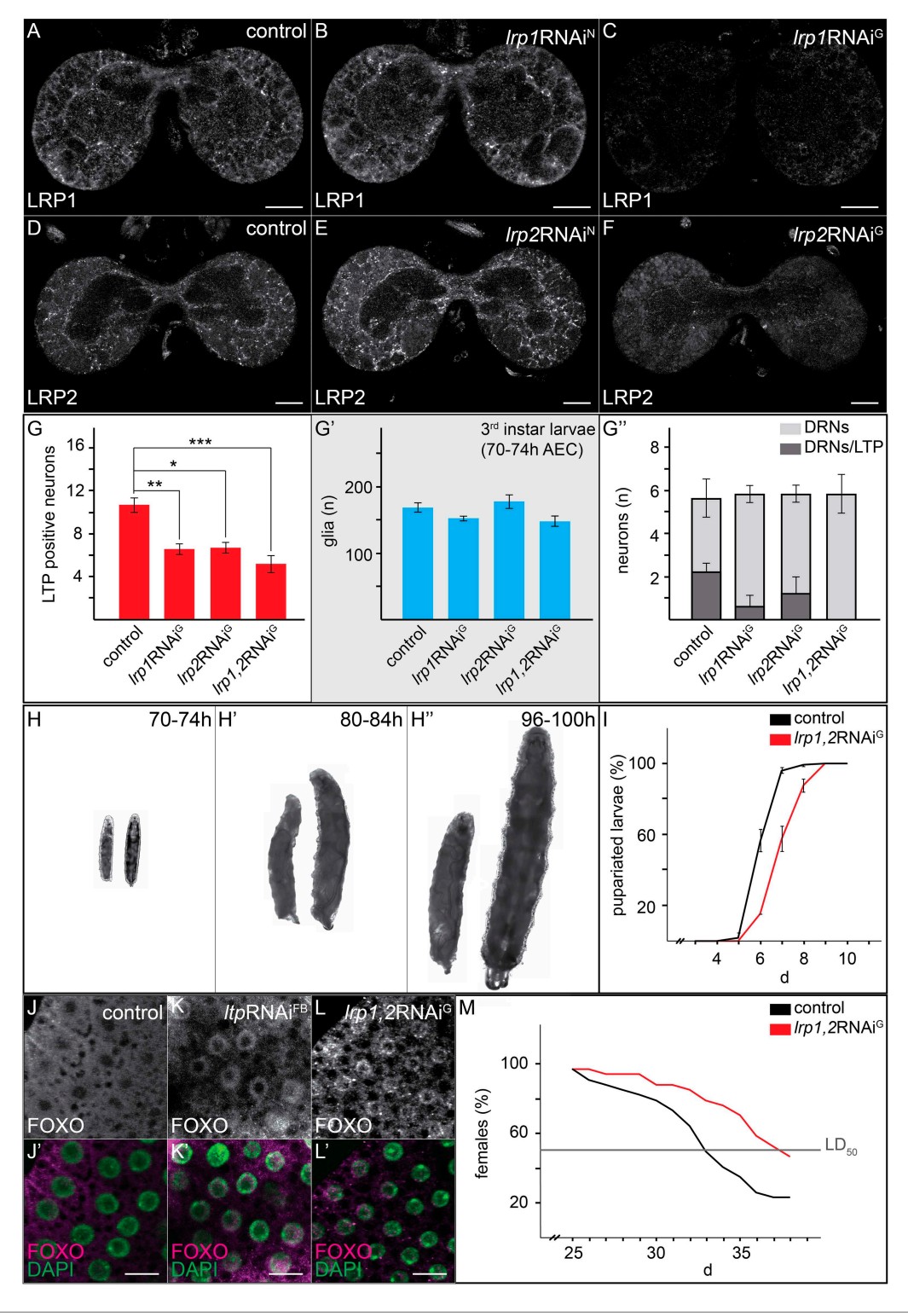

**Figure 7**. Glial LRP1 and LRP2 receptors move LTP across the BBB. (**A**–**F**) Confocal brain sections at the level of the big commissure stained for LRP1 (**A**–**C**) or LRP2 (**D**–**F**) from control[YF] larvae (**A** and **D**) or larvae with neuronal (**B** and **E**) or glial (**C** and **F**) knock-down of each receptor. Glial knock down reduces LRP1 and LRP2. (**G**–**G′′′**) Average number of LTP-positive neurons/brain lobe (**G**) Repo-positive glia/brain (**G′**) or fraction of LTP-positive DRNs (**G′′′**) in larval brains with glial knock-down of LRP1 and LRP2 singly or in combination, as indicated. Error bars indicate standard

*Figure 7. Continued on next page*

*Figure 7. Continued*

deviation. T-test significance: *p = 0.052, **p > 0.01, ***p < 0.001. (**H–H"**) control larvae (right) and larvae with glial knock-down of both LRP1/LRP2 (left) photographed at indicated times after egg collection. (**I**) Percent of control (n = 216, black) or double LRP1/LRP2 knock down (n = 183, red) larvae pupariating over time. (**J–L'**) Confocal stack projections of fat bodies from control (**J** and **J'**), fat body (**K** and **K'**) or glial (**L** and **L'**) LRP1,2 knock down larvae stained for FOXO (magenta) and DAPI (green). Scale bars = 20 μm. (**M**) Percent survival of control (n = 34, black) or glial LRP1/2 double knock-down (n = 33, red) flies fed with YF; grey line indicates 50% survival.

The following figure supplements are available for figure 7:

**Figure supplement 1**. Neuronal LTP enrichments are reduced in *lrp1,2*.

**Figure supplement 2**. Neuronal LTP enrichments are only mildly affected in *lpr1,2*.

**Figure supplement 3**. Both receptors, LRP1 and LRP2, promote LTP transport in glia.

NGS, 0.1% Triton X-100 PBS solution and antibodies were diluted as follows: anti-HA[16B12] 1:1500 (Santa Cruz Biotechnology, Dallas, TX), anti-Elav[7E8A10] 1:1500 [Developmental Studies Hybridoma Bank (DSHB), University of Iowa, Iowa City, IA], anti-LTP 1:1000 (*Palm et al., 2012*), anti-Repo[8D12] 1:1000 (DSHB), anti-LpR1[Sac8] and anti-LpR2[Sac6] (1:500), anti-LRP1 and anti-LRP2 (1:500) (*Riedel et al., 2011*), anti-Dilp2 and anti-dFOXO (1:1000, gifts from P Leopold), anti-IMPL2 (1:1000, gift from E Hafen) and DAPI 1:100000 (Roche, Germany). For quantifications, tissues were treated in parallel and imaged under identical conditions using either a Zeiss or Olympus confocal microscope. Data were analyzed using Fiji (*Schindelin et al., 2012*).

## Western blots

If not stated otherwise, larval sections were performed in ice cold PBS, samples homogenized in 1% Triton X-100 lyses buffer and proteins measured with BCA protein standard Kit (Invitrogen, manufacturer protocol, Germany). Resultant SDS-PAGE blots were probed with anti-AKT1 1:2000 (Cell Signaling, Danvers, MA), anti-ApoLII 1:4000 (*Panakova et al., 2005*), anti-CvD 1:1000, anti-LTPI 1:3000 or anti-LTPII (1:3000) (*Palm et al., 2012*).

## Fly stocks

If not stated otherwise, flies were kept at 25°C on either PF or YF under a 12hr light/12hr dark cycle. UAS:*lrp1*RNAi and UAS:*lrp2*RNAi lines are from Vienna *Drosophila* RNAi Center, and *oregonRC*, *repo*-Gal4, *dilp2*-Gal4, *dilp2*, UAS:*gfpRNAi*, UAS:*GCaMP*, UAS:*trpA1* lines are from Bloomington Stock center. Published are UAS:*ltp*RNAi, *ltp* (*Palm et al., 2012*), *imp-l2*-Gal4 (*Bader et al., 2013*), *lpp*-Gal4 (*Brankatschk and Eaton, 2010*), *lpr1*, *lpr2* (*Khaliullina et al., 2009*), *lrp1*, *lrp2* (*Riedel et al., 2011*), *moody*-Gal4 (gift from C Klaembt), *impl2*-Gal4 (gift from E Hafen). UAS:*cherry* and *rab3*-Gal4 were generated by transforming VK37 or VK33 flies (*Venken and Bellen, 2005*) with our cloned constructs. *Yrab3* and *Yrab27* are N-terminally YFP[MYC] tagged viable *rab* alleles (*rab3* and *rab27*) generated by targeted YFP[MYC] integration into the endogenous locus (publication in preparation).

## Fly food recipes

Glucose food (GF) per liter: 10 g agar, 2 g glucose, phosphate-buffered saline buffer (PBS); lipid depleted food (LDF, calculated calories = 784 kcal/l) per liter: 10 g agar, 100 g glucose, 100 g chloroform extracted yeast extract, PBS; yeast extract food supplemented with Ergosterol (LDS) per liter: 10 g agar, 100 g glucose, 100 g chloroform extracted yeast extract, 10 g Ergosterol (from a 5 mM EtOH solution), PBS; plant food (PF, calculated calories = 788 kcal/l) per liter: 10 g agar, 38 g peptone (soy), 80 g cornmeal, 80 g malt, 2 ml cold pressed sun-flower oil, 22 g treacle, 6.3 ml propionic acid, 1.5 g nipagen; yeast food (YF, calculated calories = 809 kcal/l) per liter: 10 g agar, 80 g yeast (brewers), 20 g yeast extract, 20 g peptone (soy), 30 g sucrose, glucose 60 g, 0.5 g $CaCl_2(2)H_2O$, 0.5 $MgSO_4(6)$ $H_2O$, 6.3 ml propionic acid, 1.5 g nipagen.

## Lipid extraction

40 g PF or YF were homogenized in 100 ml Chloroform. The resulting homogenates were centrifuged and the lower organic phase was removed and evaporated. The residues were re-suspended in 2 ml Chloroform and 30 μl of these solutions were applied to lipid-depleted food.

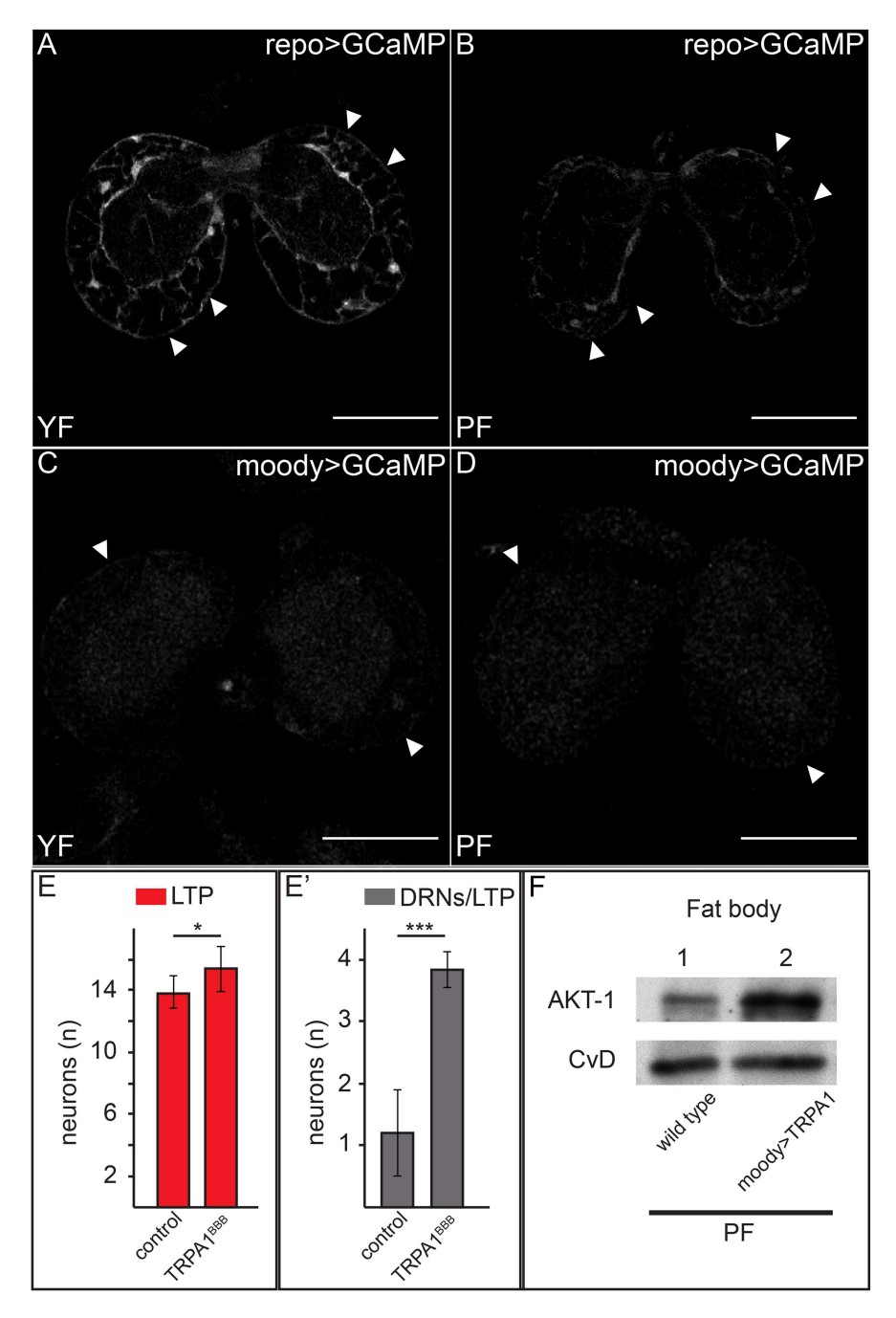

**Figure 8**. Increased free Ca++ levels promote systemic Insulin signaling. (**A–D**) Show GFP fluorescence of the GCaMP reporter construct in confocal sections of larval brains at the level of the big brain commissure. GCaMP was expressed either in all glia under the control of *repo-GAL4* (**A** and **B**) or in BBB glia under the control of *moody-GAL4* (**C** and **D**). Larvae were reared on yeast food (**A** and **C**) or plant food (**B** and **D**). White arrowheads point to blood brain barrier and scale bars indicate 50 μm. (**E**) shows the average number of LTP-positive neurons per brain lobe (**E**, red, n = 10) or the average number neurons co-staining for LTP and Dilp2 (DRNs) (**E'**, grey, n = 5) from either *UAS:trpa1* (control) larvae or larvae expressing TRPA1 under the control of *moody-GAL4* (TRPA1^BBB). All larvae were reared on plant food at 29C. Error bars show standard deviations. * indicates p < 0.02, *** indicates p < 0.0002. (**F**) Western blot of fat body lysates from control (UAS:*trpa1/+*) larvae (lane 1) or larvae expressing TRPA1 in BBB glia under the control of *moody-GAL4* (lane 2), probed for phospho-AKT1 and CvD. Larvae were all reared on plant food (PF).

## Sample collections

If not stated otherwise, embryos were collected for either 1 hr (see *Figure 4D*) or 4 hr at 25°C, and larvae raised on YF were sampled after 20–24 hr (First instar), 30–34 hr (Second instar) and 72–76 hr (early third instar). To compare PF-bred larvae to similarly staged YF-bred larvae, we staged them anatomically using mouth hook morphology. For food shift experiments, larvae were raised on YF and placed for indicated time intervals at 25°C on new food types. Larval hemolymph was prepared as described (*Brankatschk and Eaton, 2010*) and analyzed using Western blotting.

## Quantifications of neuronal staining patterns

We quantified numbers of stained cells per brain from larval CNS confocal stacks comprising the entire brain lobes but excluding the ventral ganglion. Sections were spaced 1.5 μm in Z. Within each experiment, brains were stained and recorded in parallel using the same confocal settings.

## Larval development and adult lifespan

Unless otherwise stated, embryos were collected on apple-juice-agar plates for 4 hr at 25°C and transferred to the stated diet. Newly formed pupae were counted on a daily basis. Newly hatched female flies were collected in 2–3 hr intervals at 25°C, 3 flies were placed into one vial and weighed. For lifespan experiments, females and males were kept together (2:1 ratio) at 25°C and flipped every 2 days. Deceased female animals were counted daily.

## Additional information

### Funding

| Funder | Grant reference number | Author |
|---|---|---|
| European Molecular Biology Organization | | Marko Brankatschk |
| Deutsche Forschungsgemeinschaft | NV EA24 | Marko Brankatschk, Suzanne Eaton |
| Max-Planck-Gesellschaft | | Marko Brankatschk, Sebastian Dunst, Linda Nemetschke, Suzanne Eaton |

The funders had no role in study design, data collection and interpretation, or the decision to submit the work for publication.

### Author contributions

MB, Conception and design, Acquisition of data, Analysis and interpretation of data, Drafting or revising the article; SD, Photographed some prepared CNS samples shown in (*Figure 7A–F*) and some prepared samples used to acquire quantification data, Commented on the manuscript; LN, Staged and photographed larvae shown in *Figure 4F–F'*; and *Figure 7H–H'*; Commented on the manuscript; SE, Conception and design, Analysis and interpretation of data, Drafting or revising the article

### Author ORCIDs

Marko Brankatschk, http://orcid.org/0000-0001-5274-4552

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
