## [Decision Letter]

Thank you for sending your work entitled “Delivery of circulating lipoproteins to specific neurons in the *Drosophila* brain regulates systemic Insulin signaling” for consideration at *eLife*. Your article has been favorably evaluated by Randy Schekman (Senior editor) and 3 reviewers, one of whom, Mani Ramaswami, is a member of our Board of Reviewing Editors.

The Reviewing editor and the other reviewers discussed their comments before we reached this decision, and the Reviewing editor has assembled the following comments to help you prepare a revised submission.

All the reviewers appreciated the significance, clarity and interest of the findings described in your manuscript, which link metabolism and nutrition to neuroendocrine signaling and functions via a specifically identified molecular, cell biological and neural circuitry However, a consensus from the discussion is that a few additional experiments will be necessary to support some of the key conclusions in the current paper. These, are summarized below and elaborated in the specific reviewer comments that are included with this decision letter.

We would be enthusiastic about a revised manuscript in which you:

a) Provide additional evidence to address the issue of how and why DRN neurons and IPC neurons functions are connected. In current form, there is not enough data to support a conclusion that DRNs induce IPCs to release insulin, and the role and mechanism of LTP accumulation on DRNs is unclear.

b) Try to better identify the primary nutrient signal (lipid) that stimulates LTP signaling. Some thoughtful experimental effort towards this end will be very valuable and useful, even if these prove (unfortunately) to be unsuccessful.

c) Strengthen the existing data using additional measures of insulin signaling, beyond Foxo localization.

Further detail is provided below:

Reviewer #1:

The manuscript describes a very nice series of experiments that outline the role and mechanism by which circulating lipoprotein may be transported across the BBB and regulate neurons that control the release of insulin-like peptides. LTP accumulates on a small number of neurons in the brain of which localization to a subset (the DRNs) is most tightly correlated with stimulation of insulin signaling.

The experiments are very well done and the data well presented. However, there are a few issues that should be addressed before publication.

1) Additional experiments to demonstrate or more strongly argue a causal role for DRN cells and for LTP accumulation on these neurons in promoting ILP release will greatly strengthen an important conclusion of this paper.

2) A model figure indicating the proposed pathway from nutrient signal to LTP and finally to ILP release and FOXO signaling will greatly help communicate the core model. At this point, how nutrient signal is sensed by LTP and alters the effectiveness of its transport and localization is not clear.

3) It is not obvious why DRN neurons accumulate ILPs. Is this simply an observation or is it potentially significant in a way that is consistent with LTP accumulation on these cells? Some clarification on the authors' thinking here will be useful.

4) The abbreviations and acronyms used are sometimes very distracting and occasionally make things difficult for the reader. It will be useful for the authors to either define the acronym more than once as is conventional, or take some other measures to make this easier on the non-expert reader.

Reviewer #2:

In a series of logically designed and carefully controlled experiments, Brankatschk and colleagues show that fat body-derived LTP is transported across the blood brain barrier to reach a subset of neurons. These include the insulin-producing cells regulating growth and sugar metabolism and the adjacent Imp-L2 producing cells (referred to as DRNS). The presence of LTP in neurons is regulated by nutrients and seems to modulate systemic insulin signalling, at least based on the differential subcellular localization of FOXO in the FB. This is an exciting manuscript potentially showing a novel role for a lipoprotein (LTP) in conveying nutritional information from the fly adipose tissue to the brain. I only have two major issues:

1) The involvement of the DRNS and IPCs in the described process needs to be clarified, especially with regard to signal directionality between these two neuronal populations. The authors use the correlation between LTP reduction/absence from DRNs and low systemic insulin signalling observed in their experiments (diet, FB>LTP RNAi and glial depletion of LPR1/2) to imply and sometimes conclude that LTP in DRNs is required for IPC insulin release and high insulin signalling. Based on all the data provided, would it not be equally conceivable that LTP is transported across the blood brain barrier directly to the IPCs, and the presence/absence of LTPs in DRNs is merely a correlate of insulin release from the IPCS? As the authors state, DRNS recruit insulin from IPCs using Imp-L2, so this alternative explanation seems more likely. The issue is further complicated by the fact that 1) there are other Imp-L2-expressing neurons in the VNC, including other insulin-producing neurons and 2) to my knowledge (based on Bader et al J Cell Sci), and contrary to what the authors state, direct contact between the DRNS and IPCs has not been shown.

One experiment that would begin to clarify the involvement of all these neuronal populations is to test whether expression of LTP in either DRNs (using ImpL2-Gal4, ideally compared to ImpL2Gal4, elavGal80) or IPCs (using Ilp2-Gal4) can rescue systemic insulin levels (as monitored by FB FOXO localization), increased Ilp2 production and/or release in lipid-poor conditions?

2) All the statements about systemic insulin signalling are based on the subcellular localisation of FOXO in the fat body. This is a very indirect and sometimes unclear readout, especially because LTP is produced by the FB so it could affect FOXO localisation intracellularly or in an autocrine manner. Are Ilp2 levels in IPCs or Ilp2 release affected by any of the described manipulations?

A less substantive concern is that LTP is also present in a couple of glial cells. Is that LTP also FB-derived (i.e. gone when LTP is depleted from the FB?). Otherwise it could provide a local source of CNS LTP.

Reviewer #3:

LTP (Lipid Transfer Particle) transfers lipids including fatty acids from food and lipids produced within the intestines from the intestines to a protein known as Lipophorin (LPP). Additionally LTP can aid in the transport of LPP lipids into other cells. LPP is able to cross the blood brain barrier (BBB) and accumulates within the brain. The authors link the function of LTP in the brain to nutrient sensation and systemic insulin signaling. They find a requirement for lipoproteins in mediating LTP recruitment to DRNs and that LTP function at DRNs is necessary for normally high levels of Insulin signaling. This is done by crossing the blood brain barrier. The failure to recruit LTP to these cells results in arrested development and growth as well as decreased Insulin signaling.

The authors do a very nice job showing LTP expression in distinct neurons within the brain and documenting its colocalization with specific DRNs. This paper describes an interesting link between larval diet and accumulation of LTP at these DRNs as well as being able to show that a loss of both Lrp1 and Lrp2 causes a loss of LTP within DRNs. The paper advances the current understanding of how lipid sensing is transmitted from the fat body to the brain using LRP1/2 to transport LTP across the BBB.

This paper could be strengthened by better showing some of the raw data. For example, being able to see stacks of brain sections in which all of the DRNs or LTP positive cells are in view simultaneously will support the cartoon depictions in Figure 1.

Also, FOXO nuclear/cytoplasmic localization in the fat body is pretty much all the authors use in the entire manuscript as an output of IIS within the animal. Since Insulin signaling is central to the conclusions of this manuscript, it will help to show altered insulin signaling in an additional way (phosphorylation of receptor, AKT etc.).

A central question that remains to be resolved is which lipid/s within the yeast containing food is essential for proper LTP localization. The negative nature of the plant lipids can be a clue. The authors need to supplement with combinations of lipids to the media to determine the necessary lipids for this process.

Finally, does the activation of the Insulin pathway in the ltp-RNAi background rescue the growth/life span phenotype?

---

## [Author Response]

*a) Provide additional evidence to address the issue of how and why DRN neurons and IPC neurons functions are connected. In current form, there is not enough data to support a conclusion that DRNs induce IPCs to release insulin, and the role and mechanism of LTP accumulation on DRNs is unclear*.

The effect of DRN’s on IPC neurons:

We have traced the projections of IPCs and DRNs and show that they come into very close contact in the neuropil. We have also blocked synaptic vesicle release in IMPL2-expressing neurons and shown that this slows larval growth and reduces systemic IIS as shown by lowered phosphoAKT in the fat body (Figures 5 and 6). Taken together, these data support the idea that IMPL2-expressing neurons signal to IPC’s to promote Dilp release.

The role/mechanism of LTP accumulation on DRNs:

We include new experiments showing that blocking BBB transport of LTP to DRNs reduces systemic IIS by inhibiting Dilp release by IPCs. Knock-down of the LTP transporters LRP1 and LRP2 in the BBB reduces circulating levels of Dilp2 in the hemolymph and reduces levels of phosphoAKT in the fat body (in addition to causing nuclear accumulation of Foxo). This is strong evidence that accumulation of LTP on DRNs promotes Dilp release by IPCs (Figure 5).

Our previous work showed that LRP1,2-mediated LTP transport to DRNs was required for high level systemic IIS on yeast food. We now provide the complementary result: forcing LTP accumulation on DRNs is sufficient to activate systemic IIS even when larvae are fed with plant food. We were able to do these experiments because we made the novel discovery that yeast food, but not plant food, increases cytoplasmic free calcium in blood brain barrier glial cells. Strikingly, ectopically inducing calcium influx by over-expressing the TRPA1 calcium channel in BBB glia causes LTP to accumulate specifically on DRNs even when larvae are fed with plant food. Finally, calcium influx into BBB cells is sufficient to activate systemic IIS (as reflected by phosphorylation of AKT in the fat body) even when larvae are fed with plant food. This data is now included in (Figure 8). These experiments support the idea that yeast food lipids alter Ca++ signaling in BBB glial cells, inducing them to transport LTP to DRNs, which then signal to IPCs to release Dilps.

*b) Try to better identify the primary nutrient signal (lipid) that stimulates LTP signaling. Some thoughtful experimental effort towards this end will be very valuable and useful, even if these prove (unfortunately) to be unsuccessful*.

The first step towards this goal was to show directly that the nutrient is a lipid that is present in yeast but not plant food. To do this, we prepared chloroform extracts of lipids from yeast food and from plant food and compared their ability activate IPCs (as reported by the Ca++ sensor GCaMP) and to promote larval growth. Indeed, yeast lipids activate Ca++ signaling in IPCs and speed larval growth whereas plant lipids do not. This data is now included in (Figure 5). We have already quantified the lipid composition of these foods by lipid mass spectrometry (Carvalho et al., 2012), and this suggests some intriguing possibilities; for example fatty acid chain length and unsaturation are dramatically different in these foods, and these differences directly influence the fatty acids present in all tissue lipids (including the brain). We are proceeding with experiments to precisely identify the responsible component(s) but these cannot be completed in the timeframe of this revision.

*c) Strengthen the existing data using additional measures of insulin signaling, beyond Foxo localization*.

We have now added several different readouts:

1) A calcium sensor to monitor activity of IPCs

2) Western blots of hemolymph to detect circulating Dilp2

3) Western blots to detect phospho-Akt in the fat body

4) Membrane association of PH^GFP^

Using these tools, we now clearly show that yeast food, but not plant food, activates systemic IIS by activating IPCs and promoting Dilp release.

Specifically:

1) The calcium sensor is active in IPCs of larvae fed yeast food, but not plant food (Figure 5).

2) Circulating Dilp2 levels are high in animals fed yeast food and almost undetectable in animals fed plant food (Figure 5).

3) Yeast food increases Akt1 phosphorylation in the fat body (Figure 5).

4) Yeast food increases membrane association of PH^GFP^ in the salivary gland (Figure 5).

Reviewer #1:

*The manuscript describes a very nice series of experiments that outline the role and mechanism by which circulating lipoprotein may be transported across the BBB and regulate neurons that control the release of insulin-like peptides. LTP accumulates on a small number of neurons in the brain of which localization to a subset (the DRNs) is most tightly correlated with stimulation of insulin signaling*.

The experiments are very well done and the data well presented. However, there are a few issues that should be addressed before publication.

*1) Additional experiments to demonstrate or more strongly argue a causal role for DRN cells and for LTP accumulation on these neurons in promoting ILP release will greatly strengthen an important conclusion of this paper*.

See response to consensus comment (a).

*2) A model figure indicating the proposed pathway from nutrient signal to LTP and finally to ILP release and FOXO signaling will greatly help communicate the core model. At this point, how nutrient signal is sensed by LTP and alters the effectiveness of its transport and localization is not clear*.

*3) It is not obvious why DRN neurons accumulate ILPs. Is this simply an observation or is it potentially significant in a way that is consistent with LTP accumulation on these cells? Some clarification on the authors' thinking here will be useful*.

While our data suggest that the DRNs are regulating Dilp release by IPCs, the fact that IPC-derived Dilps are found in DRNs seems to suggest that signals may pass in both directions. We now mention this in the Discussion. In mammals, insulin stimulates the activity of lipoprotein lipase, which mobilizes fatty acids from lipoproteins. Thus, one intriguing speculation is that lipids from LTP might be more effectively mobilized to DRNs when these cells receive Dilps from IPCs. Of course the IPCs likely integrate signals about many different nutrients, not only yeast lipids, to regulate Dilp release. Perhaps LTP signals to DRNs more effectively if these cells are sensitized by the Dilps derived from IPCs?

*4) The abbreviations and acronyms used are sometimes very distracting and occasionally make things difficult for the reader. It will be useful for the authors to either define the acronym more than once as is conventional, or take some other measures to make this easier on the non-expert reader*.

We now spell out fat body rather than abbreviating it as FB.

Reviewer #2:

*1) The involvement of the DRNS and IPCs in the described process needs to be clarified, especially with regard to signal directionality between these two neuronal populations. The authors use the correlation between LTP reduction/absence from DRNs and low systemic insulin signalling observed in their experiments (diet, FB>LTP RNAi and glial depletion of LPR1/2) to imply and sometimes conclude that LTP in DRNs is required for IPC insulin release and high insulin signalling. Based on all the data provided, would it not be equally conceivable that LTP is transported across the blood brain barrier directly to the IPCs, and the presence/absence of LTPs in DRNs is merely a correlate of insulin release from the IPCS? As the authors state, DRNS recruit insulin from IPCs using Imp-L2, so this alternative explanation seems more likely*.

In the new manuscript, we provide new data to support the idea that information does flow from ImpL2-producing neurons to IPCs. We would argue that LTP on DRNs is unlikely to derive from IPCs because we only rarely observe LTP on one cell of the IPC cluster under conditions where it is reproducibly found on all DRNs. If IPCs were the source of LTP on the DRNs, we might expect it to accumulate there at higher levels (like Dilps do). Of course it is hard to rule out that LTP travels so fast through the IPCs that it can’t be observed there in the steady state, but it makes the possibility seem less likely.

*The issue is further complicated by the fact that 1) there are other Imp-L2-expressing neurons in the VNC, including other insulin-producing neurons and 2) to my knowledge (based on Bader et al J Cell Sci), and contrary to what the authors state, direct contact between the DRNS and IPCs has not been shown*.

We have now demonstrated that projections from IPCs and ImpL2-expressing neurons do appear to contact each other; see response to consensus reviewer comment (a).

One experiment that would begin to clarify the involvement of all these neuronal populations is to test whether expression of LTP in either DRNs (using ImpL2-Gal4, ideally compared to ImpL2Gal4, elavGal80) or IPCs (using Ilp2-Gal4) can rescue systemic insulin levels (as monitored by FB FOXO localization), increased Ilp2 production and/or release in lipid-poor conditions?

This experiment might distinguish these possibilities if the apolipoprotein moiety of LTP, rather than a lipid provided by LTP, were responsible for its influence on IIS. But we strongly suspect that the effects are due to a nutritional lipid that is carried by LTP into the brain, so expressing LTP in neurons would not recapitulate this effect. There are also several technical difficulties in performing this experiment. LTP cannot be secreted by cells that do not express microsomal triglyceride transfer protein and it is unclear whether neurons do so. Furthermore, there are no full length cDNAs available for LTP (it is extremely large, 4333AA long) and to piece one together and make a transgenic to express it would take longer than the time frame of this revision.

2) All the statements about systemic insulin signalling are based on the subcellular localisation of FOXO in the fat body. This is a very indirect and sometimes unclear readout, especially because LTP is produced by the FB so it could affect FOXO localisation intracellularly or in an autocrine manner. Are Ilp2 levels in IPCs or Ilp2 release affected by any of the described manipulations?

We have now examined hemolymph levels of Dilp2 and shown that they are high in wild type animals on yeast food and low on plant food. Levels of circulating Dilp2 are also reduced by blocking LTP transport across the BBB using LRP1,2 knock-down. See also response to consensus reviewer comment (c).

*A less substantive concern is that LTP is also present in a couple of glial cells. Is that LTP also FB-derived (i.e. gone when LTP is depleted from the FB?). Otherwise it could provide a local source of CNS LTP*.

We showed in the previous version of the manuscript that using repoGAL4 to drive ltpRNAi does not affect the localization of LTP on neurons; only knock-down in the fat body can do this. Thus, glial cells are not the source of neuronal LTP. We didn’t describe this experiment in enough detail in the previous manuscript; in particular, we never clearly stated that fat body driven LTP knock-down reduces glial LTP as well thus, glial LTP in the brain is also derived from the fat body. We now include movies of confocal z stacks through complete brains from wild type animals and animals where *ltp* has been knocked down in the fat body (Videos 5 and 6). They are stained with anti-repo, so readers can see the reduction of LTP on glia.

Reviewer #3:

*This paper could be strengthened by better showing some of the raw data. For example, being able to see stacks of brain sections in which all of the DRNs or LTP positive cells are in view simultaneously will support the cartoon depictions in*
Figure 1.

Thanks for this suggestion. These are now available as Videos 1, 2, 3 and 4. We have also added similar movies of complete z stacks to support other statements made in the paper.

*Also, FOXO nuclear/cytoplasmic localization in the fat body is pretty much all the authors use in the entire manuscript as an output of IIS within the animal. Since Insulin signaling is central to the conclusions of this manuscript, it will help to show altered insulin signaling in an additional way (phosphorylation of receptor, AKT etc.)*.

We have now added several different readouts:

1) A calcium sensor to monitor activity of IPCs.

2) Western blots of hemolymph to detect circulating Dilp2.

3) Western blots to detect phospho-Akt in the fat body membrane association of PH^GFP^.

*A central question that remains to be resolved is which lipid/s within the yeast containing food is essential for proper LTP localization. The negative nature of the plant lipids can be a clue. The authors need to supplement with combinations of lipids to the media to determine the necessary lipids for this process*.

The first step towards this goal was to show directly that the nutrient is a lipid that is present in yeast but not plant food. To do this, we prepared chloroform extracts of lipids from yeast food and from plant food and compared their ability activate IPCs (as reported by the Ca++ sensor GCaMP) and to promote larval growth. Indeed, yeast lipids activate Ca++ signaling in IPCs and speed larval growth whereas plant lipids do not. This data is now included in (Figure 5). We have already quantified the lipid composition of these foods by lipid mass spectrometry (Carvalho et al., 2012), and this suggests some intriguing possibilities. For example, fatty acid chain length and unsaturation are dramatically different in these foods, and these differences directly influence the fatty acids present in all tissue lipids (including the brain). We are proceeding with experiments to precisely identify the responsible component(s) but these cannot be completed in the timeframe of this revision.

Finally, does the activation of the Insulin pathway in the ltp-RNAi background rescue the growth/life span phenotype?

We did not attempt this experiment because systemic loss of LTP has pleiotropic effects. In addition to reducing insulin signaling via its effects in the brain, *ltp* knock-down prevents loading of sterols onto Lipophorin into the intestine, thereby causing sterol depletion throughout the larva (Palm et al. 2012). Sterol depletion reversibly arrests larval development by a mechanism that doesn’t seem to involve insulin signaling (Carvalho et al., 2010). This was why we needed to make the more specific manipulation of blocking BBB transport of LTP to study its role on DRNs.